# Ratiometric Near-Infrared Fluorescent Probes Based on Hemicyanine Dyes Bearing Dithioacetal and Formal Residues for pH Detection in Mitochondria

**DOI:** 10.3390/molecules26072088

**Published:** 2021-04-06

**Authors:** Yunnan Yan, Yibin Zhang, Shuai Xia, Shulin Wan, Tara Vohs, Marina Tanasova, Rudy L. Luck, Haiying Liu

**Affiliations:** 1Department of Chemistry, Michigan Technological University, Houghton, MI 49931, USA; zdyunnan@163.com (Y.Y.); shuaix@mtu.edu (S.X.); swan@mtu.edu (S.W.); tavohs@mtu.edu (T.V.); mtanasov@mtu.edu (M.T.); 2College of Pharmaceutical Sciences, Gannan Medical University, Ganzhou 341000, China

**Keywords:** near-infrared fluorescence, ratiometric imaging, pH, mitochondria, fluorescent probe

## Abstract

Ratiometric near-infrared fluorescent probes (**AH^+^** and **BH^+^**) have been prepared for pH determination in mitochondria by attaching dithioacetal and formal residues onto a hemicyanine dye. The reactive formyl group on probe **BH^+^** allows for retention inside mitochondria as it can react with a protein primary amine residue to form an imine under slightly basic pH 8.0. Probes **AH^+^** and **BH^+^** display ratiometric fluorescent responses to pH changes through the protonation and deprotonaton of a hydroxy group in hemicyanine dyes with experimentally determined p*K*a values of 6.85 and 6.49, respectively. Calculated p*K*_a_ values from a variety of theoretical methods indicated that the SMD_BONDI_ method of accounting for solvent and van der Waals radii plus including a water molecule located near the site of protonation produced the closest overall agreement with the experimental values at 7.33 and 6.14 for **AH^+^** and **BH^+^** respectively.

## 1. Introduction

Mitochondria are small subcellular organelles that generate adenosine triphosphate (ATP) to power various cell functions in all eukaryotic cells [1,2,3,4]. Mitochondria also control homeostasis and redox signaling and regulate cell apoptosis and death [1,2,3,4]. An alkaline pH ≈ 8.0 is essential in mitochondria to sustain the proton motive potential during the synthesis of ATP [5,6]. Effective detections of mitochondrial pH changes provide an insightful understanding of mitochondrial physiology and pathology [7]. Many fluorescent probes have been developed for monitoring mitochondrial pH and some of these feature excellent sensitivity and high three-dimensional and temporal resolution [7,8,9]. Specific targeting of mitochondria has been achieved by employing electrostatic interactions of positively charged fluorescent probes, such as rhodamine or cyanine dyes, with the potentially negative internal membranes of mitochondria [7,8,9]. Another strategy to position dyes within mitochondria consisted of designing a hemicyanine dye to covalently link with mitochondrial proteins through a direct displacement of the reactive chlorine group on the fluorophore [10].

It is well-known that the formyl group easily reacts with primary amines to form imine derivatives as Schiff bases [11,12,13,14,15]. For this reason, we developed a reactive ratiometric near-infrared fluorescent probe (**BH^+^**) for pH detection in mitochondria by attaching a formyl group to a hemicyanine dye in order to prevent the probe from diffusing away from mitochondria. We also contrast results with a ratiometric fluorescent probe bearing a thioacetal residue (**AH^+^**), which could be hydrolyzed in cells, resulting in covalent linking to proteins. Ratiometric near-infrared fluorescent probes have desirable advantages such as near-infrared imaging to achieve deep tissue penetration, low photodamage to cells, and less biological sample autofluorescence. Our probes also possess self-calibration capability with two emissions to overcome systematic errors of intensity-based fluorescent probes produced by excitation light fluctuation, probe concentration changes, uneven distribution, and compartmental localization [16,17,18,19,20,21,22,23,24]. Both **AH^+^** and **BH^+^** probes (Scheme 1) show ratiometric fluorescent probes to pH changes with the largest bathochromic shifts of 38 nm and 59 nm based on the protonation and deprotonation of a hydroxy group attached to the hemicyanine dyes, respectively.

## 2. Results and Discussion

### 2.1. Probe Design and Synthesis

The fact that formyl and dithioacetal groups chemically react with amine residues has been widely used for the covalent modification of proteins. We sought to utilize this property to prevent the fluorescent probe from diffusing out of mitochondria and designed probes **AH^+^** and **BH^+^** that carry dithioacetal and formyl moieties, respectively, on the hemicyanine dye. We envisioned that the formyl moiety could directly interact with proteins upon localization of the probe in mitochondria. On the other hand, the dithioacetal could hydrolyze to yield a reactive formyl group in cells for an analogous covalent linking with proteins. The chemical synthesis of these probes started with protecting an aldehyde group of 2,4-dihydroxybenzaldehyde (**1**) by converting it into a dithioacetal residue, yielding 4-(1,3-dithiolan-2-yl)benzene-1,3-diol (**2**). Reacting compound **2** with cyanine dye (IR-780) (**3**) in DMF under basic conditions generated a hemicyanine dye bearing a dithioacetal residue (probe **AH^+^**). Deprotection of the dithioacetal residue in probe **AH^+^** under an acidic environment produced the hemicyanine dye bearing a formyl group (probe **BH^+^**) (Scheme 2). 

### 2.2. Optical Responses of Probes **AH^+^** and **BH^+^** to pH Changes

UV-vis studies of probes **AH^+^** and **BH^+^** show that both probes are sensitive to changes in pH level. At pH 4.0, probe **AH^+^** shows two absorption peaks at 611 nm and 658 nm (Figure 1, left). A gradual increase of pH from 4.0 to 10.1 causes a significant red shift in absorption with a broad absorption peak appearing at 700 nm. The observed shift can be attributed to the stabilization of a negatively charged hemicyanine portion of the dye due to the deprotonation of the hydroxyl group on the hemicyanine dye under basic conditions. Probe **BH^+^** exhibits a main absorption peak at 586 nm, and a shoulder peak at 630 nm at pH 4.0 (Figure 1, right). Gradual increases in pH result in considerable red shifts to two absorption peaks at 630 nm and 684 nm at pH 10.1. The changes in absorption for probe **BH^+^** can be also observed visually as the color of the solution changes from purple to blueish green with increasing pH (Figure 2).

Standard pH titrations of the probes were also conducted to assess fluorescence changes. At pH 4.0, probe **AH^+^** shows a fluorescence peak at 680 nm at pH 4.0 (Figure 3, left). Gradual increase of pH from 4.0 to 10.1 results in decreased fluorescence emission at 680 nm and increased emission at 718 nm with an overall bathochromic shift of 38 nm (Figure 3, left). The experimental p*K*_a_ value for **AH^+^** is 6.85 (Figure 4). The quantum yields measured for **AH^+^** at pH 4.1 and 9.2 are 0.06% and 0.27%, respectively, (Table 1). Alterations in solution pH also impacted fluorescence for probe **BH^+^**. At pH 4.0, probe **BH^+^** displays a fluorescence peak at 667 nm. Changing the pH of the solution from 4.0 to 10.1 induced a ratiometric red shift in fluorescence emission to 715 nm, with an overall bathochromic shift of 48 nm. The experimental p*K*_a_ value for **BH^+^** is 6.49 (Figure 4). In comparison to **AH^+^**, probe **BH^+^** is slightly blue-shifted in fluorescence due to the electron-withdrawing nature of the formyl group (Figure 3). The quantum yields measured for probe **BH^+^** at pH 4.1 and 9.2 are 1.9% and 12.3%, respectively (Table 1). It is noteworthy that both probes behave as fluorescent pH sensors as is evident from the experimentally-observed reversible red or blue shifts due to the reversible protonation and deprotonation of the hemicyanine dye upon pH changes as the probes show satisfactory reversibility between pH 10.0 and pH 4.0 (Figure 5). 

### 2.3. Theoretical Results

We further employed computational models to assess the pH-induced geometric changes and the nature of the electronic transitions, and to calculate p*K*_a_ values for both probes. For **A**, **AH^+^**, **B,** and **BH^+^**, stable converged geometries in almost completely planar conformations of the pseudo-rhodamine and hemicyanine moieties were obtained (Appendix A). This planarity resulted in a complete delocalization of the π-orbitals, with the apparent energy transition (Figure 6) between protonated and deprotonated forms. In all cases, this main transition consisted of the delocalization of electronic density from LCAO-HOMO to LCAO-LUMO upon deprotonation of the hydroxyl moiety. We found the calculated (Appendix A) and experimental absorption values to be in good agreement for **B** (0.18 eV) and **BH^+^** (0.23 eV) probes (within the expected 0.20–0.25 eV range [25]) but they were off for **A** (0.29 eV) and **AH^+^** (0.35 eV) probes. Calculations also identified H-bonding between the dithioacetal and hydroxyl functionalities for **AH^+^** that could contribute to the appearance of the red-shifted transition for **AH^+^** at low pH (700 nm shoulder, Figure 1) and also influence the equilibrium between the protonated and deprotonated OH group. Analogous H-bonding between the carbonyl and OH, although not detected through calculations, could contribute to the shoulder in the UV absorption of **BH^+^**. 

Given the plethora of functional and basis set combinations available today, calculating p*K*_a_ values represents a challenging task. However, recent publications have suggested a more directed route to accomplish these calculations. The nature of the calculation involves getting the values listed in Equation (1), where *G**_aq_ refers to the calculated standard free energies of the deprotonated and protonated species in solution and *G**_aq_(H**^+^**) = −270.30 kcal/mol [26]. The bases for the nature of the calculation have been described previously for carboxylic acids, amines, and thiols [27].
(1)$$pK{a\;=\;[G}_{aq}^{*}{(A}^{-}{)\;-\;G}_{aq}^{*}{(\mathrm{AH}}^{+}{)\;+\;G}_{aq}^{*}{(H}^{+})]\;/\;(2.303\;x\;\mathrm{RT})$$

In order to derive the p*K*_a_ values of alcohols, in addition to the utilization of the SMD method [28], it was suggested to model an H_2_O adduct hydrogen-bonded to the alcohol (Figure 7) [29]. We conducted calculations on these models and the data are summarized in Table 2. The calculations based on the IEF-PCM (the inclusion of a dielectric medium) model of aqueous solvation afforded a reasonable p*K*_a_ = 6.37 value for probe **AH^+^** that is comparable to the experimental value of 6.85. However, for probe **BH^+^**, a significant difference in values (0.264 calculated vs. 6.40 experimental) was observed.

Based on previous reports, we performed calculations using the SMD variant of solvation (i.e., intrinsic Coulomb radii), including SMD_Bondi_ (intermolecular van der Waals radii [32]) and SMD_sSAS_ (a scaled solvent-accessible surface) [27]. As listed in Table 2, p*K*_a_ values of 8.52, 8.54, and 9.64 were calculated for probe **AH^+^** for the three methods, respectively. Using the same methods, the p*K*_a_ values of 6.48, 5.81, and 7.60 were calculated for the **BH^+^** probe. In this case, a close correlation between the calculated and experimental (6.49) p*K*_a_ values was observed for the **BH^+^** probe but not for the **AH^+^** probe. Considering the shared H-bonding between the hydroxyl and dithioacetal moieties calculated for **AH^+^**, it is plausible that the discrepancy in p*K*_a_ values for **AH^+^** arise due to the large errors in calculating the p*K*_a_ values (~6 p*K*_a_ units) for thiols using this method [27]. As a next step, we calculated p*K*_a_ values after including a hydrogen-bonded water molecule near the hydroxyl group (Figure 7) [29]. Similar to previous calculations, a good agreement with experimental data was observed for **BH^+^** (calculated p*K*_a_ values of 6.27, 6.14, and 6.92). For **AH^+^**, a reasonable correlation with the experimental p*K*_a_ value was observed using the SMD_Bondi_ method (Table 2, Bondi(H_2_O)). This agreement may pertain to the additional H-bonding between the added water molecule and a lone pair on the hydroxyl’s O atom (Appendix A), which would essentially mimic the partially-solvated OH bond within **AH^+^**. Overall, the SMD_Bondi_ method, with one water molecule included, was able to provide a close correlation with experimental p*K*_a_ values, owing to the inclusion of a solvation factor. This would suggest that this type of model could serve as a useful starting point for theoretical p*K*a calculations on structurally similar and related molecules. 

### 2.4. Assessing H^+^ Specificity of Probes **AH^+^** and **BH^+^**

In order to evaluate the probe selectivity to pH over other potential interferents, the fluorescence spectra of the probes were recorded in the presence of various cations, anions, and amino acids at the physiological conditions (pH 7.4). We found that cations such as K**^+^**, Mg^2**+**^, Al^3**+**^, Ba^2**+**^, Fe^3**+**^, Co^2**+**^, Ni^2**+**^, Sn^4**+**^, Cu^2**+**^, Zn^2**+**^, Cd^2**+**^, Hg^2**+**^, Mn^2**+**^, Cr^3**+**^, Pb^2**+**^, and Fe^2**+**^ (Figure 8), and anions such as Cl^−^, CO_3_^2−^, HCO_3_^−^, SO_4_^2−^, SO_3_^2−^, HSO_3_^−^, NO_3_^−^, PO_4_^3−^ and S_2_O_3_^2−^ do not cause any significant fluorescence changes in probe fluorescence (Figure 9). Probe fluorescence was also independent of the presence of amino acids (DL-cysteine, DL-homocysteine, DL-alanine, DL-arginine, DL-leucine, DL-tyrosine, glutamic acid, glycine) and glutathione (GSH) (Figure 10). These results confirmed that the probes possess good selectivity to pH and suggested that probes are useful as pH sensors in a complex biological environment. 

### 2.5. Photostability of the Probes

We then performed the photostability experiment of these two probes. As shown in Figure 11, the intensity of probes did not show significant changes either under an acid environment, at about pH 4.0, or at neutral pH 7.0, indicating that the probe shows excellent photostability (Figure 11). 

### 2.6. Cell Cytotoxicity

We evaluated the cell cytotoxicity of the probes for their biocompatibility by MTT assay. The cytotoxicity of the probe increases slightly with the probe concentration with lower cell viability. High concentrations (50 µM) of the probe do not cause considerable cytotoxicity because the cell viability is still higher than 91%, indicating that the probe shows excellent biocompatibility and low toxicity (Figure 12). IC_50_ values of probes **AH^+^** and **BH^+^** are 135 μM and 155 μM, respectively, when the cell viability rate reaches 50% after incubation of HeLa cells with probe **AH^+^** or **BH^+^**.

### 2.7. Analysis of Mitochondrial Localization for **AH^+^** and **BH^+^** Probes in Live Cells

To assess the validity of our assumption that probes **AH^+^** and **BH^+^** can specifically accumulate inside of mitochondria [33,34], we conducted colocalization studies in live cells, using a commercial available mitochondria-specific dye, i.e., Mitoview Blue, for the control. As probes **AH^+^** and **BH^+^** exhibit near-infrared fluorescence (>667 nm), it is easy to distinguish their fluorescent signal from biological background fluorescence as that occurs at a lower wavelength (i.e., 400–500 nm). For live-cell imaging, HeLa cells were incubated with **AH^+^/**Mitoview Blue and **BH^+^/**Mitoview Blue mixtures. The dye-treated cells were further taken for confocal imaging. Imaging analysis has shown that both **AH^+^** and **BH^+^** probes have effectively penetrated the cellular membrane and localized in the cytoplasm. **AH^+^** and **BH^+^** probes were visible using a 635 nm excitation/700–750 nm emission channel (channel II, Figure 13). Mitoview Blue was visible in these cells under a 405 nm excitation and 425–475 nm emission (channel III, Figure 13). Colocalization analysis of the probes **AH^+^** and **BH^+^** with Mitoview Blue resulted in Pearson’s correlation coefficient values of 0.924 and 0.955, which confirms that the new probes are localized within mitochondria. 

We further assessed whether the presence of the electrophilic formyl or dithioacetal groups contributes to the retention of these probes in mitochondria. It has been well established that the membrane potential is the driving force for the uptake of fluorescent lipophilic cations in mitochondria [35]. It has also been shown that ionophores can effectively prevent the uptake of such species [35]. Furthermore, changing the membrane potential has also been shown to promote the efflux of charge species from mitochondria [36]. On these bases, it is expected that changing the membrane potential in probe-treated HeLa cells would induce the efflux of fluorescent lipophilic cations from mitochondria. Therefore, we sought to alter the mitochondrial membrane potential to assess the retention of either probe **AH^+^** or **BH^+^** within the mitochondria. We used FCCP (carbonyl cyanide p-(tri-fluoromethoxy)phenyl-hydrazone)—the ionophore demonstrated to uncouple oxidative phosphorylation in mitochondria, disrupt the proton gradient along the inner mitochondrial membrane, and acidify mitochondria [35].

HeLa cells were first incubated with **AH^+^** and **BH^+^** and then treated with FCCP (carbonyl cyanide p-(tri-fluoromethoxy)phenyl-hydrazone) [37,38,39]. After the subsequent treatments, we used fluorescence microscopy to assess the results (Figure 14 and Figure 15). Before the FCCP treatment, only near-infrared fluorescence in channel II (700–750 nm) was observable for **AH^+^** and **BH^+^** probes, corresponding to the existence of these probes in a non-protonated state. After FCCP treatment, a new fluorescence signal became observable in channel I (650–675 nm). The intensity of this new signal increased as fluorescence in channel II decreased. The appearance of a new fluorescence signal appears to be informative of probe protonation and stabilization of more blue-shifted **AH^+^** or **BH^+^** and indicative of mitochondrial acidification. We also observed that FCCP appears to induce a background green fluorescence in cells (Figure 14 and Figure 15). The green fluorescence co-localizes with the fluorescence induced by **AH^+^** and **BH^+^** probes, suggesting the same site accumulation. The observed lack of **AH^+^** or **BH^+^** diffusing from the mitochondria supports the idea that the introduced electrophilic formyl and dithioacetal moieties reacted and bonded with interstitial mitochondrial proteins for the observed mitochondrial retention. 

### 2.8. Assessing **AH^+^** and **BH^+^** as Intercellular pH Sensors

In order to demonstrate that our probes can respond to intracellular pH changes, we incubated HeLa cells with **AH^+^** and **BH^+^**, adjusted intracellular pH to 9.0, 8.0, 7.5, 6.5, 6.0, and 5.0 by nigericin (H**^+^**/K**^+^** ionophore) [16,18,23,24,40,41,42,43,44], and collected cellular fluorescence in two channels from 650 nm to 657 nm, and from 700 nm to 750 nm under excitation of 635 nm. With the confocal fluorescence imaging (Figure 16), upon gradual decrease of intracellular pH from 9.0 to 5.0, we observed an expected gradual decrease in fluorescence in the channel II and gradual increase in fluorescence in the channel I (Figure 16 and Figure 17). Merged images of channels I and II show color change from deep red to green, validating **AH^+^** and **BH^+^** as ratiometric reporters of intracellular pH changes (Figure 16).

## 3. Experimental Section

### 3.1. Computational Analysis of Probes A, **AH^+^**, B, and **BH^+^**

Models for probes **A**, **AH^+^**, **B**, and **BH^+^** were generated using previously published procedures [38] and conducted with density functional theory (DFT), using the APFD functional [45] and electron basis sets initially at the 6-31g(d) level to convergence in Gaussian 16 [46]. The results from this level were refined in a Polarizable Continuum Model (PCM) of water [47] with 6-311^+^g(d) basis sets, and frequency calculations were conducted. Imaginary frequencies were not obtained. Seven excited states were assessed using TD-DFT optimizations [48] in a Polarizable Continuum Model (PCM) [48] in water with the 6-311^+^g(d) basis set. p*K*a calculations were conducted as described previously with the 6-311^+^g(d) basis set with the optimized SMD implicit solvation method as described on AQUA-MER [26,27] on the converged models. p*K*a calculations were also conducted with the models enclosing one additional water molecule H-bonded to the active -OH group [29]. Harmonic frequencies were derived in all cases to ensure that the structures obtained (atomic coordinates listed in Appendix A) were minima on the potential energy surface and to obtain the required thermodynamic data for the p*K*_a_ calculations. Results were interpreted using GaussView 6 [30] for all data and figures and are presented as Supporting Information.

### 3.2. Reagents and Methods

All solvents were purchased from Sigma-Aldrich (Saint Louis, MI, USA) and used directly. All solvents for spectroscopic studies were HPLC grade without fluorescent impurities and water was deionized. Dulbecco’s modified Eagle’s medium (DMEM) high glucose with stable glutamine with sodium pyruvate, Dulbecco’s phosphate-buffered saline (DPBS), fetal bovine serum (FBS), trypsin 0.25%-EDTA in HBSS, and penicillin-streptomycin were purchased from Aldrich-Sigma. Thiazolyl Blue Tetrazolium Bromide (MTT) was purchased from Sigma-Aldrich and used as received without further purification. ^1^H and ^13^C NMR spectra were recorded at room temperature with a Varian Unity Inova 400 MHz spectrometer. All spectra of the probes were recorded in pH buffer containing 30% (*v*/*v*) ethanol. The following abbreviations are used to indicate the multiplicity: s—singlet; d—doublet; t—triplet; q—quartet; m—multiplet. The coupling constants are expressed in Hertz (Hz). Cary 60 UV-Vis spectrometer and Jobin Yvon Fluoromax-4 spectrofluorometer were used for absorption and fluorescence spectra, respectively. Confocal images were taken with Olympus FluoViewTM FV1000 using the FluoView software. Fluorescence imaging was done with an EVOS FLAuto inverted microscope.

### 3.3. Cell Culture and Cytotoxicity Assay

HeLa cells were cultured in the high glucose DMEM media supplemented with 10% FBS, 1% penicillin streptomycin in an incubator (37 °C, 5% CO_2_) [16,18,23,24,40,41,42,43,44]. Cells were replenished with fresh medium every 2 days. The cellular metabolic activity was detected using colorimetric MTT assay, reflecting the loss of metabolic activity through the reduction of tetrazole (MTT) to formazan in mitochondria. For the assay, Hela cells were seeded in 96-well flat-bottomed plates at 1 × 10^4^ cells per well and allowed to adhere for 24 h in the incubator (37 °C, 5% CO_2_). The medium was then removed, and a probe was added to the cells at different concentrations. The culture medium was added to the control group. Cells were then incubated (37 °C, 5% CO_2_) for 6 h, the probe solution was removed, and cells were rinsed with DPBS gently. MTT reagent (0.5 mg/mL) in culture media was then added to the individual wells and incubated (37 °C, 5% CO_2_) for 3 h. The MTT reagent was then removed from the cells, and DMSO (100 µL/well) was added to dissolve formazan crystals. The plate was gently shaken using a mixer to solubilize the crystal for 30 min. The absorbance was read at 550 nm using a Spectra Max i3x microplate reader. The data points from each concentration of probe were obtained from three wells and all experiments were repeated three times.

### 3.4. Cellular Imaging

Hela cells were seeded on glass coverslips of a confocal dish without surface treatment and allowed to adhere for 24 h. Subsequently, the medium was removed and the glass coverslips rinsed gently with DPBS. The cells were then stained with organelle trackers including commercial Mitoview Blue for 30 min in DPBS solution, before staining with the probe. The organelle markers are susceptible to potential oxidases in serum, therefore we used DPBS instead of a complete culture medium. Indeed, when we used a complete culture medium, mitochondria were not stained by the markers. After the cells were stained with one of the organelle trackers, the cells were then sequentially stained with the probe for 90 min in an incubator (37 °C, 5% CO_2_) and the fluorescence images were obtained by a confocal microscope every 10 min. The cells were visualized simultaneously in each channel by a confocal fluorescence microscope, image analysis, and the data of colocalization between probe and organelle trackers was obtained by using ImageJ software.

## 4. Conclusions

Two ratiometric near-infrared fluorescent probes (**AH^+^** and **BH^+^**) based on hemicyanine dyes bearing thioether and formal residues have been developed for mitochondrial pH detection in live cells. The probes show reversible and ratiometric fluorescent responses to pH changes based on the protonation and deprotonation of a hydroxy group in hemicyanine dyes. Probes **AH^+^** and **BH^+^** possessed experimental p*K*_a_ values of 6.85 and 6.49, respectively, which are close to calculated p*K*_a_ values of 7.33 and 6.14 for **AH^+^** and **BH^+^**.

## Data Availability

Not applicable.

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
