# Peer review of "Ratiometric Near-Infrared Fluorescent Probes Based on Hemicyanine Dyes Bearing Dithioacetal and Formal Residues for pH Detection in Mitochondria"

_molecules, 2021, doi:10.3390/molecules26072088_

Round 1

Reviewer 1 Report

The author's group, constituted by a nice stable research group, show the design, synthesis and application of two pH-sensitive molecules to be used as mitochondrile markers capable of functioning as a detector for mitocondrial pH. The work is well constructed and follows the line already traced in other publications (even more complex) by the authors.

In fact, in some cases the work seems a bit repetitive compared to other their previous works. The progress and the difference with respect to previous works are not perceived. This point should be improved.

Furthermore, I have not found any information regarding the synthesis and characterization of the compounds (intermediates and final products). Have they already been published in other works? where is it possible to find NMR spectra, mass, menting point, ir spectra or other information?

It would be interesting to know the target mitochondrial receptor for these molecules. It is difficult to think that it is only a non-specific localization.

minor fixes:

scheme 2: TFH? possibly THF

figure 13: correct the names of the molecules A or AH+ ? B or BH+?

Figures 13, 14 and 15: correctly report the excitation wavelengths (630 or 635?) In the figures and captions of the figures

Line 243 (figure 9), probably is a mistake. 

Based on these mancanz eritengo that work must be reviewed before publication. 

Author Response

We addressed the second reviewer’s comments as follows:

  • I have not found any information regarding the synthesis and characterization of the compounds (intermediates and final products). Have they already been published in other works? where is it possible to find NMR spectra, mass, menting point, ir spectra or other information?

We addressed this issue as the first reviewer also raised it.  We added NMR spectra and high-resolution MS data (Figures S14-S19) in supporting information.

  • It would be interesting to know the target mitochondrial receptor for these molecules. It is difficult to think that it is only a non-specific localization.

We conducted co-localization experiments by using Mitoview Blue and show that colocalization analysis of the probes AH+ and BH+ with Mitoview Blue resulted in Pearson’s correlation coefficient values of 0.924 and 0.955, which confirms that the new probes are localized within mitochondria.  

  • scheme 2: TFH? possibly THF

We corrected this typo in scheme 2.

  • figure 13: correct the names of the molecules A or AH+ ? B or BH+?

We corrected that by using probes AH+ and BH+.

  • Figures 13, 14 and 15: correctly report the excitation wavelengths (630 or 635?) In the figures and captions of the figures

Excitation wavelength is 635 nm in Figures 13, 14  and 15.

  • Line 243 (figure 9), probably is a mistake. 

Figure 9. Fluorescence intensities of 10 μM probes AH+ (left) and probe BH+ (right) in the absence and presence of different anions (200 μM) in pH 7.4 buffers under excitation at 635 nm, respectively.

We appreciate the reviewers’ comments and thank you very much for further reviewing our manuscript for publication in Molecules.

Yours sincerely,

Haiying Liu

Professor, Ph.D.

Department of Chemistry

Michigan Technological University

Houghton, MI 49931

Phone: 906-487-3451

Reviewer 2 Report

Yan et al. reported two cyanine-based probes for mitochondrial pH detection. The manuscript shall be further considered after addressing the following comments. 

  1. Line 38~40, the reactivity of formyl group and thioacetal group in cellular environment should be backed up with references.
  2. There is no structural characterization reported in the manuscript for these two probes (AH+, BH+). The NMR and MS results are required to be added. 
  3. What is the concentration of the compound in Figure 2, 3, and 5?
  4. Please elaborate what "cyclic index" in Figure 5 means, and add detailed explanation on what Figure 5 is evaluating and indicating. The current sentence around Line 90~91 is far from clear. 
  5. Error bars from Figure 8~10 are quite narrow and uniform. The authors are suggested to provide the original value of each biological replicate in the supporting information.
  6. Figure 11 should also be provided with replicates. Why was it done in 10% ethanolic solution instead of regular PBS? Please add the explanation in the manuscript. 
  7. Cytotoxicity at a fixed concentration is not reflecting the biocompatibility range. Please provide the IC50 data of these two probes.
  8. Scale bars were not labeled or captioned in Figure 13~16.
  9. The effect of Oligomycin-treatment should also be provided in Figure 14, as the Oligomycin-treatment and FCCP-treatment changes the membrane potential in opposite directions. Concentration and time length of FCCP- and Oligomycin-treatment should also be provided.
  10. In Figure 15, what is the quantitative scale of the ratio image? How was the image generated? Please include the information. 
  11. The authors should add the incubation/treatment time in each of the figure captions throughout the manuscript. 
  12. Were the imaged live or after fixation? Please add the details. 
  13. Two references on the mitochondrial targeting of cyanine dyes should be included: DOI: 10.1093/nar/gkq050; DOI: 10.1021/acs.bioconjchem.0c00079. 
  14. The authors are claiming the ratiometric sensing properties but did not provide any ratiometric analysis in the most critical figure - Figure 16. Please provide the quantitative analysis. 

Author Response

We addressed the first reviewer’s comments as follows:

  • Line 38~40, the reactivity of formyl group and thioacetal group in cellular environment should be backed up with references.

We cited references 11-15 for this issue.

  • There is no structural characterization reported in the manuscript for these two probes (AH+, BH+). The NMR and MS results are required to be added.

We added NMR spectra and high-resolution MS data (Figures S14-S19) in supporting information.

  • What is the concentration of the compound in Figure 2, 3, and 5?

We added missing concentration information (10 µM) of probes AH+ and BH+ in Figures 2, 3 and 5.

  • Please elaborate what "cyclic index" in Figure 5 means, and add detailed explanation on what Figure 5 is evaluating and indicating. The current sentence around Line 90~91 is far from clear.

We changed “cyclic index” and replaced by “cyclic pH change number” in Figure 5, and “as the probes show satisfactory reversibility between pH 10.0 and pH 4.0 (Figure 5)” to the manuscript.  We also caption of Figure 5 as “Figure 5. Fluorescence ratio changes of 10 mM probes AH+ (left) and BH+ (right) versus cyclic pH changes from 4.0 to 10.0 or 10.0 to 4.0.”

  • Error bars from Figure 8~10 are quite narrow and uniform. The authors are suggested to provide the original value of each biological replicate in the supporting information.

We provided related data in Tables S38-S43 in supporting information. 

  • Figure 11 should also be provided with replicates. Why was it done in 10% ethanolic solution instead of regular PBS? Please add the explanation in the manuscript.

We provided related data in Tables S37-S37.  We conducted this experiment in PBS solutions containing 1.0% ethanol.  “10% ethanol” is a typo.  We changed Figure caption as “Figure 11.  Photostability of 5 μM fluorescent probe AH+ and BH+ at pH 4.0 and pH 7.4 buffer solutions containing 1.0% ethanol with three-repeated experiments every 30 minutes.”

  • Cytotoxicity at a fixed concentration is not reflecting the biocompatibility range. Please provide the IC50 data of these two probes.

We add IC50 data of the probes.  We added this sentence “ IC50 values of probes AH+ and BH+ are 135 µM and 155 µM, respectively when the cell viability rate reaches 50% after incubation of HeLa cells with probe AH+ or BH+.” to the manuscript.

  • Scale bars were not labeled or captioned in Figure 13~16.

We added scale bar number to Figures 13-16.

  • The effect of Oligomycin-treatment should also be provided in Figure 14, as the Oligomycin-treatment and FCCP-treatment changes the membrane potential in opposite directions. Concentration and time length of FCCP- and Oligomycin-treatment should also be provided.

We added related references 37-39.

  • In Figure 15, what is the quantitative scale of the ratio image? How was the image generated? Please include the information.

Ratio image was obtained by using ImageJ software, which is free from https://imagej.nih.gov/ij/.

  • The authors should add the incubation/treatment time in each of the figure captions throughout the manuscript. Were the imaged live or after fixation? Please add the details.

We have added the detailed information about incubation and treatment time to the figure captions.

Figure 13. Fluorescence images of HeLa cells incubated with 10 µM probes AH+ (left), BH+ (right), and 5.0 µM Mitoview Blue. The first and second channels in the fluorescence images were recorded from 650 nm to 675 nm, and from 700 nm to 750 nm under excitation at 635 nm, respectively.  The third channel in the images was collected from 425 nm to 460 nm under excitation of 405 nm. Scale bars: 10 µm.  HeLa cells were incubated with 10 µM probe AH+ or BH+ in media at 37 oC for 20 min.  Live cell images were obtained by an Olympus IX 81 confocal microscope.

Figure 14. Fluorescence images of HeLa cells incubated with 10 µM probes AH+ before and after FCCP treat-ment. The first and second channels in the fluorescence images were recorded from 650 nm to 675 nm, and from 700 nm to 750 nm under excitation at 635 nm, respectively. Scale bars: 10 µm. HeLa cells were incubated with 10 μM probe  AH+ for 20 min, and further treated with 10 μM FCCP in PBS 37 oC for 20 min. Live cell images were obtained by an Olympus IX 81 confocal microscope.  ImageJ software was used to obtain ratiometric images. The software is free from https://imagej.nih.gov/ij/.

Figure 15. Fluorescence images of HeLa cells incubated with 10 µM probes BH+ before and after FCCP treat-ment. The first and second channels in the fluorescence images were recorded from 650 nm to 675 nm, and from 700 nm to 750 nm under excitation at 635 nm, respectively. Scale bars: 10 µm.  HeLa cells were incubated with 10 μM probe BH+ for 20 min, and further treated with 10 μM FCCP in PBS at 37 oC for 20 min. Live cell images were obtained by an Olympus IX 81 confocal microscope. ImageJ software was used to obtain ratiometric im-ages. The software is free from https://imagej.nih.gov/ij/. 

Figure 16. Fluorescence images of HeLa cells incubated with 10 µM probes AH+ (left) and BH+ (right) and nigericin to homogenize the intracellular pH of the cells with the surrounding medium at different pH values from 9.0, 8.0, 7.5, 6.5, 6.0, to 5.0. The first and second channels in the fluorescence images were recorded from 650 nm to 675 nm, and from 700 nm to 750 nm under excitation at 635 nm, respectively.  HeLa cells were incubated with 10 µM probe AH+ or BH+ for 20 min first, and then treated with 5 µg/mL nigericin in different pH buffer solutions at 37 oC for 20 min. Live cell images were obtained by an Olympus IX 81 confocal microscope.

  • Were the imaged live or after fixation? Please add the details.

“Live cell images were obtained by an Olympus IX 81 confocal microscope” was added to Figures 13-14.

  • Two references on the mitochondrial targeting of cyanine dyes should be included: DOI: 1093/nar/gkq050; DOI: 10.1021/acs.bioconjchem.0c00079.

We add these references 34 and 35 as suggested.

  • The authors are claiming the ratiometric sensing properties but did not provide any ratiometric analysis in the most critical figure - Figure 16. Please provide the quantitative analysis.

We added Figure 17 for fluorescence intensity of two channels to the manuscript. We modified the sentence as follows “In order to demonstrate that our probes can respond to intracellular pH changes, we incubated HeLa cells with AH+ and BH+, adjusted intracellular pH to 9.0, 8.0, 7.5, 6.5, 6.0, and 5.0 by nigericin (H+/K+ ionophore), [16, 18, 23, 24, 40-44] and collected cellular fluorescence in two channels from 650 nm  to 657 nm,  and from 700 nm to 750 nm under excitation of 635 nm.”

We appreciate the reviewers’ comments and thank you very much for further reviewing our manuscript for publication in Molecules.

Yours sincerely,

Haiying Liu

Professor, Ph.D.

Department of Chemistry

Michigan Technological University

Houghton, MI 49931

Phone: 906-487-3451

Round 2

Reviewer 1 Report

the corrections are exaustive for the publication

Reviewer 2 Report

The authors have properly addressed the comments from previous reviewers.